# Investigating the Feasibility of a Restaurant Delivery Service to Improve Food Security among College Students Experiencing Marginal Food Security, a Head-to-Head Trial with Grocery Store Gift Cards

**DOI:** 10.3390/ijerph18189680

**Published:** 2021-09-14

**Authors:** Ryan J. Gamba, Lana Mariko Wood, Adianez Ampil, Alina Engelman, Juleen Lam, Michael T. Schmeltz, Maria M. Pritchard, Joshua Kier Adrian Santillan, Esteban S. Rivera, Nancy Ortiz, Darice Ingram, Kate Cheyne, Sarah Taylor

**Affiliations:** 1Department of Public Health, California State University, East Bay, Hayward, CA 94542, USA; alina.engelman@csueastbay.edu (A.E.); juleen.lam@csueastbay.edu (J.L.); michael.schmeltz@csueastbay.edu (M.T.S.); mpritchard4@horizon.csueastbay.edu (M.M.P.); jsantillan@horizon.csueastbay.edu (J.K.A.S.); erivera14@horizon.csueastbay.edu (E.S.R.); nancyortizmph@gmail.com (N.O.); 2University Libraries, California State University, East Bay, Hayward, CA 94542, USA; lana.wood@csueastbay.edu; 3Pioneers for H.O.P.E., California State University, East Bay, Hayward, CA 94542, USA; aampil@horizon.csueastbay.edu (A.A.); darice.ingram@csueastbay.edu (D.I.); 4Department of Research, Alameda County Community Food Bank, Oakland, CA 94621, USA; kcheyne@accfb.org; 5Department of Social Work, California State University, East Bay, Hayward, CA 94542, USA; sarah.taylor@csueastbay.edu

**Keywords:** college students, marginal food security, food insecurity, supplemental nutrition assistance program, restaurant delivery service, food aid preferences, food assistance intervention

## Abstract

Restaurant delivery services have gained in popularity among college students; however, students participating in the Supplemental Nutrition Assistance Program (SNAP) are not allowed to redeem their benefits via restaurant delivery services. This mixed-methods head-to-head crossover trial assessed whether college students experiencing marginal food security prefer benefits via a grocery store gift card (as a proxy for traditional SNAP benefits) or via a restaurant delivery service gift card of equivalent value, and which type of benefit is more effective at improving food security status. Thirty college students experiencing marginal food security were recruited to receive $80 in cash equivalent benefits to spend over a two-month period in the form of grocery store gift cards and restaurant delivery service gift cards. Participants completed surveys and interviews to measure their food security status and share their experiences with each benefit type. After four months of benefits, 48.3% of participants improved their food security status. However, neither type of benefit was statistically better at improving food security status. Most participants preferred grocery store benefits (89.7%) over restaurant delivery service benefits (10.3%). However, more research is needed to explore whether allowing SNAP recipients to redeem their benefits with restaurant delivery services is a viable mechanism to address food challenges among college students experiencing marginal food security.

## 1. Introduction

Marginal food security exists “when an individual has problems at times, or anxiety about, accessing adequate food, but the quality, variety, and quantity of their food intake were not substantially reduced” [1]. A report from the California State University (CSU) system found the prevalence of marginal food security to be 22.4% among college students in the CSU system [2]. Although the United States Department of Agriculture systematically combines those with no food challenges with those who are experiencing marginal food security into its definition of a food secure group [3], evidence suggests that those experiencing marginal food security often have worse physical and mental health outcomes than those without food challenges [4]. Among college students, marginal food security has been positively associated with lower grade point averages, less cooking and food agency, worse perceived health, increased body mass index, and depression [5,6,7,8].

College students are eligible for the U.S. Supplemental Nutrition Assistance Program (SNAP, formerly referred to as food stamps), the largest federally funded food assistance program. Program participation has been positively associated with food security status in adults [9]. Participation rates among college students are very low; while it is estimated that 18% of college students are eligible for the program, just 3% participate [10]. College students must meet additional SNAP eligibility criteria, and this additional criteria is not clearly laid out or communicated with SNAP agencies [11]. Due to the COVID-19 pandemic, temporary policies were implemented to change SNAP eligibility criteria, resulting in 5 million additional college students becoming temporarily eligible for SNAP at the beginning of 2021 [12]. However, it is unclear how many of these newly eligible college students will pursue program participation. Barriers to SNAP participation for college students are well documented and include the onerous application process, the complicated eligibility criteria, and social stigma [13].

Some states, including California, participate in the SNAP Restaurant Meals Program (RMP), which allows homeless, elderly, and disabled populations to use their SNAP benefits at restaurants [14,15]. However, RMP does not allow participants to redeem benefits with restaurant delivery services. Restaurant delivery services have become more popular among college students [16,17] and they may be a viable mechanism to improve the food security status of college students experiencing marginal food security because such services often eliminate the need to procure, cook, or store food [5,6,18,19]. Past research suggests that some students struggle with procuring food due to lack of transportation or disability, some are uncomfortable with their culinary skills or might not have all the cooking tools they need, and some lack space to store food [5,6,18,19]. Students experiencing marginal food security might be more amenable to restaurant delivery services as knowing they have prepared food might diminish any anxiety they have surrounding the procurement or preparation of food [3]. Those who are food insecure and facing more dire food situations may also value having the food prepared and delivered. However, the cost associated with this convenience limits the amount of food they could receive and thus may push them to prefer traditional SNAP benefits. Despite the potential of restaurant delivery services, we were unable to identify a study that assessed how this type of service may be used to improve food security status.

To determine if restaurant delivery services may be a tool to address food insecurity among college students, we have identified two aims. First, we aim to assess whether college students experiencing marginal food security will have a greater improvement in food security if they receive benefits via a restaurant delivery service application available on their phones compared to grocery store gift cards (as a proxy for traditional SNAP benefits) of equivalent value. Second, we aim to determine if college students experiencing marginal food security would prefer to receive benefits via a restaurant delivery service or a grocery store gift card. To investigate these aims, we conducted a mixed-methods head-to-head crossover trial where participants received both grocery store and restaurant delivery service benefits over two months; and at the end of the four months, we assessed their preference and food security status. We hypothesize that students will prefer receiving benefits via the restaurant delivery service and that it will be more beneficial in addressing food insecurity given the convenience of not having to procure, cook, or store food.

## 2. Materials and Methods

### 2.1. Participants and Design

We recruited current students at the California State University, East Bay (CSUEB) campus who were ≥18 years old, had a smartphone, and were experiencing marginal food security. This study was limited to students experiencing marginal food security because there is little published research on interventions for this group, and this group is more likely than students experiencing low or very low food security to shift to food secure as a result of a small benefit amount [5]. A smartphone was included in the eligibility criteria to ensure participants could fully take advantage of the restaurant delivery service by ordering on their phone. Recruitment was done across campus via printed flyers and email through student listservs. Thirty students met eligibility criteria and participated in this study between June and November 2019. After written informed consent was collected, students were provided an online survey to collect basic demographic information and interviewed on campus about their experiences with marginal food security. The results of that interview are published elsewhere [5]. Participants then received $40 per month for two months in the form of a Grubhub (restaurant delivery service) gift certificate available through their smartphones, and $40 per month for two months in the form of a gift card to a grocery store of their choice. Students were enrolled on a rolling basis as we instituted block randomization to ensure that for every two participants qualifying for this study, one would be randomized to receive their restaurant delivery benefits first, and the other the grocery store benefits (Figure 1). While a single college student can receive up to $204 per month in SNAP benefits [12], amounting to approximately $3 per meal [20], the amount of $40 per month, per participant was determined in part by ethical considerations about undue coercion and also intended to ensure participants could purchase a minimum of two meals with the restaurant delivery service benefit. Upon completing each two-month period, participants completed an online survey that assessed their food security status. There were also several questions in this survey to capture their experiences with each food benefit immediately following their experience. Upon completion of both intervention types, in-person semi-structured interviews were conducted, and each participant was asked questions about their experiences with the food assistance methods. Participants also chose what type of benefits they would like to receive for their final benefit ($40) for their 5th month.

### 2.2. Measurement

Food security status was measured with the United States Department of Agriculture’s 10-item Adult Food Security Survey Module (AFSSM) [21]. While there is no scale specifically designed to assess food insecurity for college students [22], the AFSSM is frequently used to measure college students’ food security status [23]. The AFSSM includes questions about progressively more severe food situations. To be classified as marginally food secure, participants must answer affirmatively to at least one, but no more than two, questions in the AFSSM. Individuals experiencing marginal food security most frequently respond affirmatively to one or two of the first three questions from the AFSSM: (1) “I worried whether (my/our) food would run out before I got money to buy more”; (2) “The food that I bought just didn’t last, and I didn’t have money to get more”; (3) “I couldn’t afford to eat balanced meals”. of the responses “often true” and “sometimes true” (as opposed to “never true”) are considered affirmative responses. Immediately following each two-month intervention, participants were asked to fill out the AFSSM for the previous month.

### 2.3. Quantitative Analysis

A 2 × 2 contingency table was created to assess whether either type of food assistance was more adept at improving food security status (improved/not improved). A participant’s food security status was considered to improve if it changed from their baseline status of marginal food security to food secure and considered to not improve if it remained marginally food secure or got worse and they became food insecure. McNemar’s chi-squared test was implemented to determine whether either method of food assistance resulted in a greater proportion of participants improving their food security status. A Wilcoxon signed-rank test was used to assess if the food security status of all participants improved after the four-month intervention. This non-parametric test was implemented because one participant had their food security status worsen to food insecurity after the intervention, so food security was considered an ordinal outcome of food secure, marginally food secure, and food insecure. A one-sample test of proportion was implemented with STATA’s prtest command (STATA SE Version 5, StataCorp LLC, College Station, TX, USA) to determine whether there was a statistically significant difference in the proportion of participants who preferred Grubhub gift cards or grocery store gift cards. STATA SE Version 5 was used for all quantitative analysis.

### 2.4. Qualitative Analysis

In-person interviews were conducted and audio-recorded by two experienced interviewers. The interview script (Appendix A) was created to assess how participants were able to use their gift cards, the challenges they experienced, the strengths and weaknesses of each type of benefit and to probe for stories regarding their experiences with each benefit. The initial script was pilot tested among our student researchers and research team. Four researchers transcribed audio recordings word for word, and would add notes about the interview if there were background noises, inaudible conversations, etc. Interviews were analyzed for thematic content using a general inductive approach [24]. The thematic analysis approach is flexible and straightforward, the steps include familiarizing yourself with the data, generating initial codes, augmenting and coming to agreement about the codes, identifying the initial themes, and after reviewing the themes naming and defining them [24]. After an initial qualitative analysis training overview with an experienced qualitative researcher, our research team met once or twice a week to discuss issues and questions about coding during the coding process. The coding decisions were tracked and noted in the meeting notes to create an audit trail to refer to. The first three transcripts were coded as a group as a pilot test to standardize the coding process. Subsequent interviews were coded separately by at least three researchers to encourage agreement and standardization. Differences in coding were reconciled through an iterative process of review and discussion. All coding was conducted in Dedoose (version 8.3.17, Dedoose, Hermosa Beach, CA, USA).

## 3. Results

Participants in this study were predominantly undergraduate females aged 18–24 (Table 1). Almost everyone indicated they had access to cooking equipment, but 27.6% indicated they did not have adequate room to store food. More than half worked at least part time (>0 h/week) and half of the participants were unsure if they were eligible for SNAP (Table 1).

One participant was lost to follow up after receiving one month of grocery store benefits, and therefore was not included in this study. Two participants did not come to collect their grocery store gift cards, thus missing their second benefit, but were included in this study because excluding these participants did not alter the statistical significance of our findings. Two participants filled out their food security questionnaire after two months of intervention, but not after four months, and therefore they were excluded from analysis requiring their four-month food security status. One participant consistently declined to answer questions, and therefore we created a “missing” category. Approximately 30% of participants reported they had leftover benefits at the end of their two months of benefits and this was distributed fairly evenly across both benefits (Table 2). However, most participants described having small amounts that they could not use without using some of their own money. Almost three of every four participants (73.3%) indicated this was their first time using this particular restaurant delivery service. There were seven instances where four unique participants shared their benefits with someone else, meaning a participant allowed a non-participant to use an electronic gift card from this study and/or a participant shared the food they purchased with benefits with non-participants at some point in this study.

Results from the Wilcoxon signed-rank test determined there was a statistically significant difference in food security status before and after the four months of benefits, where food security status significantly improved in 14/29 participants and got worse in 1/29 participants (z = 3.36, *p* ≤ 0.001). We compared participants’ food security status after each intervention with their marginal food security status at baseline. Of those who received grocery store benefits, 12/28 (42.9%) improved their food security status from marginally food secure to food secure. Of those who received Grubhub benefits, 14/28 (50%) improved their food security status from marginally food secure to food secure. Six of the 27 (22.22%) of those that had food security measurements after each intervention improved food security after both interventions. Results from the McNemar’s chi-squared test found that neither type of benefit was statistically better at improving food security status (*p* < 0.10). The one-sample test of proportion found that a significantly (*p* < 0.001) higher proportion of participants preferred the grocery store gift card for their last benefit (89.7%, n = 26) compared to the restaurant delivery service (10.3%, n = 3).

Participants mentioned in interviews how the interventions improved their food security status and overall well-being, often citing the reduction in stress, the ability to make healthier food choices, and improvements in going to the gym and studying. One participant remarked, “...just being able to buy healthier foods, so I was better able to function throughout the day to study”. Most participants discussed specifically how the grocery store benefits made a positive impact on their food situation. A participant discussed the additional funds allowing them to make healthy choices when stating, “Well I think I could afford (healthy food), but it was more like there were times where I’m like I’m doubling guessing, like “do I really need it, like this is similar” I guess. And kind of knowing that like I had that security or that money is just for healthy food, made me feel like 100% positive about the decision I guess. Cause sometimes I felt like I’m being wasteful by buying myself something healthy”.

Table 3 describes the benefits and challenges described by the participants for each method of food assistance. Respondent interviews took 27.7 min on average. Participants consistently mentioned receiving more food with the grocery store benefit and having healthier options as key advantages of that method. Other advantages mentioned included being in control of every ingredient going into their body, their ability to budget, finding sales/good deals to get more food, buying food that lasts longer, having more options, and getting to buy food they would enjoy cooking with; convenience was mentioned by some who lived close to their grocery stores. For the restaurant delivery service, participants consistently discussed the convenience of the time and energy saved by not having to go to a store and the benefit of not having to cook. Some participants also mentioned they appreciated getting to choose from a variety of places and getting to see the food they are going to buy, the security of knowing they have money for food, and the freedom to buy food that they might not otherwise purchase. While very few participants discussed challenges with the grocery store gift card, some mentioned it was not as convenient as having someone deliver food to them. There were more challenges associated with the restaurant delivery service, with the most commonly cited challenges being the poor food quality, issues with pricing including tipping, fees, and taxes, and, most consistently, getting less food for the benefits they were given. Other challenges include the delivery being delayed, food options being limited in certain areas, and food order cancellations with slow refund times.

## 4. Discussion

In contrast with our hypothesis, when given the option to select their final (5th) monthly benefit, participants preferred the grocery store benefit over the restaurant delivery service. Interviews with participants revealed that the increased amount of food procured with the grocery store benefit was the deciding factor in participants’ preferences. However, 10.3% (three participants) did wish to receive their final $40 of benefits from the restaurant delivery service. These three participants cited the convenience and the time saving of not having to cook or travel as reasons for their choice. Further research among SNAP eligible non-participants is needed to determine how participation in SNAP may change if benefits could be redeemed via a restaurant delivery service.

Both interventions significantly improved the food security of college students experiencing marginal food security. Assessments of food security interventions on campus are lacking leaving comparisons difficult. Campus food pantries are often evaluated. However, they tend to focus on participation and dietary outcomes [25,26]. Limited research on campus gardens suggests that student turnover, limited financial support, and unclear policies and regulations make it an unreliable source to improve the food security status of college students [27]. While meal swipe programs and text messaging systems to make college students aware of leftover food on campus have also taken hold on college campuses [28,29] we could not identify any studies that have investigated changes in food security status promoted by these interventions.

While we were unable to identify studies that investigated how interventions improved the food security status of college students, our results mostly contrast with findings from recent food insecurity interventions designed to improve the food security status of children. Interventions including $40 a month in SNAP benefits [30], boxes of food worth $37 plus $15 of vouchers to purchase fruits and vegetables [31], and from $1–$122 of SNAP benefits based on income and distance to the nearest grocery store [32] were all found to have no significant impacts on the children’s food security status. The intervention that did improve food security status for school-aged children is the Summer Electronic Benefits Transfer for Children pilot program which provided $60 or $30 per child, per month during the summer months [33]. Our difference in results may be because we studied college students who have more autonomy than children to address their food security, and because those experiencing marginal food security may need much smaller benefits to improve their food security status than those who are food insecure. Additionally, we did not use a true control group, and some of the participants may have changed their food security status due to external circumstances without receiving any benefit.

While we hypothesized that college students experiencing marginal food security might be more greatly assisted by a prepared meal, we found no significant difference in food security improvement between the two interventions. Additionally, because students experiencing marginal food security might not be going without food, we anticipated that for some students their stress would be better addressed by knowing that they can have a prepared meal when they need it as opposed to more benefits to a grocery store. While a prepared meal that is delivered to them may do more to alleviate this particular stress than the grocery benefit, we may not have seen greater improvement in food security after using the restaurant delivery service because worrying about food is just one component of the myriad struggles faced by those experiencing marginal food security. Respondents experiencing marginal food security typically answered affirmatively to 1–2 of the first three questions of the food security module, which include worrying about running out of food and not having money to buy more, the food not lasting and not having money to buy more, and not eating balanced meals. The latter two may largely not be addressed by the $40 a month (typically two meals) purchased with the restaurant delivery benefit. In particular, participants often discussed how there were not many nutritious options when using the restaurant delivery service, perhaps not helping them feel as though they were eating balanced meals. Additionally, participants had more challenges using the restaurant delivery service, and the security of knowing they had a prepared meal they could order on their phone at any time was eroded by several barriers. These barriers included a high proportion using this app for the first time, limited food options to choose from, and problems with the delivery process, including slow delivery times and incorrect orders. Considering all of the challenges reported with the restaurant delivery benefits, it is worth reiterating that both methods significantly improved food security status to the same extent.

There are several strengths of this study. This analysis was the first to assess how benefits for a restaurant delivery service may impact food security status in direct comparison with a restaurant delivery app. In contrast to other studies focusing on the amount of food, this study examines whether the mechanism by which people with food challenges receive food assistance affects their food security status. This study adds to the small but growing literature focusing on college students experiencing marginal food security. Additionally, this study examined a racially diverse population and reduced the potential for confounding by implementing a randomized head-to-head crossover trial.

There are important limitations of this study. All of the participants were from a single university, which limits generalizability. Participants redeeming benefits may not have used the benefits entirely for themselves, as there were seven reported instances where participants shared their benefits with others. Relatedly, we did not procure information regarding participants’ living situations, which may have provided additional context about sharing food and their experiences with each intervention type. There were issues with physical gift cards, and in redeeming electronic benefits via the restaurant delivery service which could have impacted how participants perceived their benefits. In particular, as many participants used this restaurant delivery service for the first time and experienced challenges, this likely lowered their appreciation for the benefits received via the restaurant delivery service. Despite randomization of which benefits were received first, some participants indicated that how much they appreciated the benefits might have been impacted had they received benefits in a different order. They indicated this was not due to the lack of a washout period but because some participants received some of their benefits during summer when their schedules were different from the academic year. While it was not our initial plan to run part of this study over the summer, it was made necessary because of when funding was received. This study used a grocery store gift card rather than grocery delivery services, because, at the time of this study, grocery delivery services were not available in many geographical areas in which our participants resided. The small sample size coupled with the proportion of those that improved with one method of food assistance and not the other (51.85%) suggests that we would have needed to observe a difference in the proportion of those who improved their food security status from each method by about 35% to detect a statistically significant difference at *p* = 0.05 and 0.8 power. However, our findings that the proportion of those who improved food security status were very similar making this limitation less of a concern.

## 5. Conclusions

In conclusion, $80 of benefits over two months significantly improves the food security status of college students experiencing marginal food security, regardless of whether these benefits are redeemed via a restaurant delivery service or grocery store. Because only 10% (3 participants) preferred receiving benefits from a restaurant delivery service, more research is needed to determine if meal delivery is a viable option to serve college students experiencing marginal food security.

## Figures and Tables

**Figure 1 ijerph-18-09680-f001:**
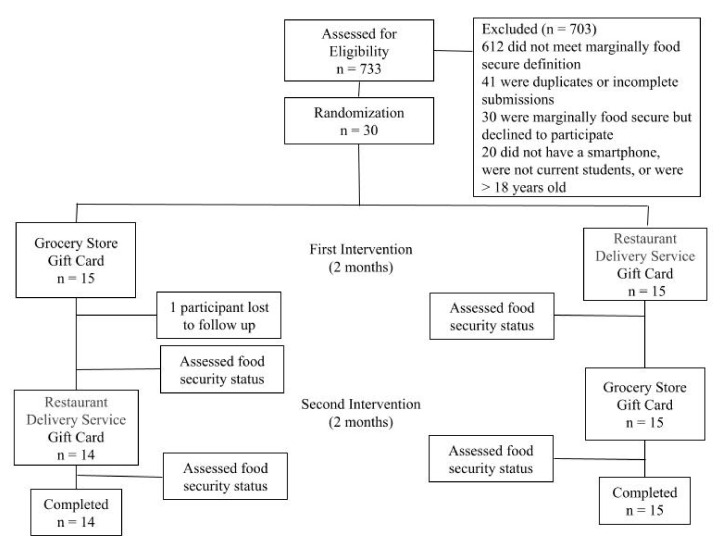
Research study timeline.

**Table 1 ijerph-18-09680-t001:** Characteristics of marginally food-secure students as California State University, East Bay (n = 29).

Characteristics	Selection	n (%)
Age (in years)	18–24	22 (75.86)
	25–29	4 (13.79)
	≥30	2 (6.90)
	Missing	1 (3.45)
Gender	Female	22 (75.86)
	Male	6 (20.70)
	Missing	1 (3.45)
Race/Ethnicity ^1^	Asian	22 (75.86)
	Hispanic/Latinx	6 (20.70)
	White/Caucasian	1 (3.45)
	Filipinx/Pacific Islander	22 (75.86)
	African American/Black	6 (20.70)
	Indian	1 (3.45)
	Middle Eastern	22 (75.86)
	Missing	6 (20.70)
Transfer vs. Native Students	Transfer	16 (55.17)
	Native	12 (41.40)
	Missing	1 (3.45)
Graduate vs. Undergraduate Students	Undergraduate	23 (79.31)
	Graduate	5 (17.24)
	Missing	1 (3.45)
Hours Worked per Week	0	5 (17.24)
	1–≤10	4 (13.79)
	11–≤20	8 (27.59)
	21–≤30	5 (17.24)
	>30	4 (13.79)
	Seasonal	1 (3.45)
	Missing	2 (6.90)
Financial Support from Parents/Caregivers/Others	Receiving Support	18 (62.07)
	Not Receiving Support	10 (34.48)
	Missing	1 (3.45)
Eligibility to Participate in SNAP ^2^	Unsure of Eligibility	15 (51.72)
	Eligible	6 (20.70)
	Not Eligible	8 (27.59)
Adequate Food Storage	Yes	22 (75.86)
	No	8 (27.59)
Adequate Cooking Equipment	Yes	27 (93.10)
	No	1 (3.45)
	Missing	2 (6.90)

^1^ The participants do not add up to 29 because one student selected multiple races. ^2^ U.S. Supplemental Nutrition Assistance Program (SNAP, formerly referred to as food stamps).

**Table 2 ijerph-18-09680-t002:** Food benefit issues and preferences (n = 29).

Issue	Selection	n (%)
Selected Grocery Store Gift Card for Final (5th) Month?	Yes	26 (89.66)
	No	3 (10.34)
Used All Benefits ^1^	Yes	14 (48.28)
	No	15 (51.72)
Received New Food Assistance in the Last 60 Days	Yes	4 (13.79)
	No	24 (82.76)
	Missing	1 (3.33)
First Time Using Grubhub	Yes	22 (73.33)
	No	7 (23.33)
	Missing	1 (3.33)
Timing of Benefits Mattered ^2^	Yes	16 (53.33)
	No	12 (40.00)
	Unclear	1 (3.33)
Shared Benefits	Shared Grocery Store Benefits	3 (10.00)
	Shared Grubhub Benefits	4 (13.33)
	Shared Grocery Store and Grubhub Benefits	3 (10.00) ^3^
	Did Not Share Either Benefit	25 (83.33)

¹ Participants who did not exhaust 100% of their allotted benefits are marked as “No”; often they had a small amount remaining and were unable to purchase anything with the remaining balance. ^2^ Participants said that the timing of when they received each intervention may have affected how they appreciated the benefits. ^3^ Three of the same participants shared both the grocery store and meal delivery service benefit while one participant only shared the meal delivery service benefit.

**Table 3 ijerph-18-09680-t003:** Benefits and disadvantages of grocery store and restaurant delivery benefits, themes from participant interviews.

Categories	Themes	Quotes
Grocery Store Benefit Themes	Greater Purchasing Power	“You got way more food, and then you can kind of tailor your meal to what you want, same as you know going grocery shopping so you don’t have to order off so and so’s menu and then be like, well I’m gonna get this because this is what I like and then take off x, y, z or add x, y, z for more money. It was just kind of like, well I can already make that and just buy out things but not only I can make it and buy all the things but I’ll have that same meal multiple times instead of just once so you got more multiple meals for probably like the same amount of money.” “With groceries. I felt like with the grocery you can buy a lot of food and it can even last for weeks.”
	Greater Options and More Healthy Options	“Of course, at a grocery store, you get more, and it’s healthier, you could really pick what you want.” “With the grocery store, I liked it more because it was more customizable in the sense that you really get to pick what you want, you could span out breakfast, lunch, dinner, it goes farther, and of course it was healthier and allowed me to kind of figure out what I wanted to eat for that week instead of just picking whatever was on (restaurant delivery service), because there is no like- not every store in Hayward is on (restaurant delivery service), so you have a very few selected restaurants where you could choose from.”
Restaurant Delivery Service Benefit Themes	Not Having to Cook	“Two biggest benefits is that it’s very not as time-consuming as like you cooking and like, for example, a student needs time to study and do academics in which I could just open up my laptop, order it, and keep studying while I wait for my food to get there and then just have like a mini-break to eat my food and get back to studying which was really convenient times-wise.” “The fact the food was coming to you was obviously a huge benefit and then you don’t have to spend any time preparing that food. It was just like, alright thanks and you can go eat.” “I didn’t have to cook, I didn’t have to clean my pots and pans.”
	Convenience of Not Having to Travel	“I liked that it was, like I didn’t have to make food. I kinda just had to put in an order and then I could probably have more time to do homework or something while it was delivered to my door.” “I think the convenience to be able to get a meal without having to necessarily like spend as much effort for going out to grocery shop or going to a restaurant, and being able to spend that time on other things.”
Restaurant Delivery Service Challenges Themes	Pricing/Tipping	“Not a lot of options and expensive.” Interviewer: “Okay, so you say for like GrubHub, you generally sacrificed part of your like health? Or like the healthier option?” RG25: “Yeah. Because it’s just the, okay you want, you’re like you want this one but since it’ll cost you more and then you’ll just like oh, change your mind and then just go to what you can afford.” “Yeah like a service fee or something and then the delivery, and then the delivery, and then the tip for the driver, and like cuz it’s my first time using it so the first time which I kind of feel bad for doing this but the first time it kinda just it automatically keeps doing this where like it calculates how much you spend on food and like how much you have to pay the driver like either 10%, 20%, 40%, and then after that like the website calculates it by itself and it tells you how much you’re supposed to pay and you just like click pay now and that’s it, you know the order has been sent but the second time I kinda started playing around with the website and I actually found like this button where it says that you can like put another tip like another amount of tip and you can like change it to where it automatically sets it up to and then instead of like for example if I paid like $10 for like a plate or something and instead of paying the driver like $3 or $4 or whatever, I would like delete it and put like 50 cents or something so like I would have more money.”
	Poor Food Quality	“I feel like it’s just temptation of just like you know, it’s a lot of fast food, so, you know obviously there’s a lot like, not a lot of healthy options I guess.” “Um, Let’s see… Well, in the beginning I was searching for healthier options for Grubhub, but then like I said, out of the stores here in the area, I was able to find one store, but the food wasn’t that appetizing.”
	Cancelations/ Delays	“Yeah, this was just this one time. But, one thing that was consistent is that they always took forever. That was something, was something that was consistent, yeah.” “Yeah, so there was like twice I tried ordering and they canceled my food order on me, so it was just kind of like—it’s not—I just think it’s the app, if the app was built better, like that was my only complaint, if it was like made better, like to work better, it would have been okay.”

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
