# Peer review of "Investigating the Feasibility of a Restaurant Delivery Service to Improve Food Security among College Students Experiencing Marginal Food Security, a Head-to-Head Trial with Grocery Store Gift Cards"

_ijerph, 2021, doi:10.3390/ijerph18189680_

Round 1

Reviewer 1 Report

This article seeks to understand if grocery store gift cards or restaurant delivery gift cards are better at improving food insecurity and which is preferred among a sample of 29 college students.

First, I have serious concerns about the framing of the article, which could pretty easily be revised. The current set-up of the article is unrealistic and should be removed. The authors suggest that SNAP benefits may be distributed via a grocery card (EBT) or as a gift card for restaurants (i.e., to receive hot prepared food).  The problem is that SNAP does not allow recipients to use funds at restaurants (delivery or in-person). There is no serious policy discussion considering this change, and in my opinion, it will not happen anytime soon.  Instead of the SNAP framing which doesn’t make sense, I encourage the authors to frame this as an option for non-profit institutions, like colleges or food pantries, to consider. These institutions have flexibility to buy and distribute grocery cards or restaurant delivery gift cards.  (My point is further emphasized by the fact that only 6/29 participants reported that they are eligible for CalFresh (CA SNAP – which should be explained for the non Cali reader, BTW)).  As such, this paper may be a better fit for a higher education audience, rather than a public health audience.

Pg 1 line 35: The abstract emphasizes some students like restaurant delivery:  “Some participants (10.3%) preferred redeeming benefits via the restaurant delivery service, suggesting participation in SNAP may increase among college students if this were allowed.”
But this statement is very misleading for several reasons.

First, the results indicate “when given the option to select their final monthly benefit, participants preferred the grocery store benefit over the restaurant delivery service” (Pg 10 line 257).
--Why isn’t that the takeaway in the abstract? You could revise it to say something like:  Most students prefer grocery store benefits over restaurant delivery service, though three students preferred restaurant delivery.  If non-profits/colleges want to be inclusive, they could consider giving students with marginal food security an option of which gift card they prefer, since both appear to improve food security at similar rates.

Second: In prior research, students report that they don’t take up SNAP because they’re not aware of it, don’t think they’re eligible, or have trouble submitting the forms and going through the bureaucracy to actually obtain it. There is no evidence indicating that college students aren’t signing up for SNAP because it only offers grocery benefits.  And the study here does not provide evidence on that hypothesis either.  Suggesting that students don’t have SNAP because they want restaurant vouchers is not only absurd, frankly, it’s dangerous.  

The methods section should income a power analysis, or reference it. How was the total size of 30 determined? What kind of an effect size would have been needed to detect a difference between treatments? What literature was used to estimate the potential effect size?

Was there baseline equivalence between the groups? The pre-treatment background characteristics must be shown for each group.  I also recommend that covariates are included in impact analyses to increase precision. For a guidebook on how to present RCTs, I recommend the What Works Clearinghouse from the U.S. Department of Education. Given the topic of the study, the authors should be adept at speaking with a higher education audience as well as a public health audience.

Pg. 2 line 55: “College students are eligible for [SNAP].”   Yes, this is true, but it omits a really important part of the story.  That is, college students have to meet an additional criterion above and beyond standard income, asset, and household guidelines. 

Author Response

Thank you for your review and helping to make our article stronger. Please see our responses to your specific comments attached.

Reviewer 2 Report

Abstract

Line 28: Since the goal seems to be to test whether being able to use SNAP benefits for restaurant delivery would be beneficial, I suggest changing what’s in the parentheses to “as a proxy for traditional SNAP benefits.”

Line 33: You may want to consider changing “discuss” to something such as “share their experiences with”

Lines 35-37: This conclusion doesn’t seem to match the results. As mentioned in my comments on the conclusion of the paper overall, suggesting future research on this rather than policy changes seems more appropriate.

Introduction

Lines 66-67: It would be helpful to mention what SNAP benefits can be used for. You may want to reference California’s Restaurant Meals Program.

Line 80: I suggest changing “probably pushing” to “may push”

Line 87: Since the goal seems to be to test whether being able to use SNAP benefits for restaurant delivery would be beneficial, I suggest changing what’s in the parentheses to “as a proxy for traditional SNAP benefits.”

Methods

Lines 114-116: It would be helpful to mention how this amount compares to what a typical or average benefit for college students receiving SNAP would be.

Lines 122-123: Was this a fifth month of benefits?

Mention somewhere in the participants and design section when the data were collected (months and years). It’s important context to know whether the data were collected before or during the COVID-19 pandemic and during the school year and/or summer months.

It may also be helpful to provide additional justification as to how the two conditions were selected. There are two differences between the conditions – delivery vs. obtaining food in-person and restaurant food vs. groceries. I’m wondering why grocery delivery vs. restaurant delivery or gift cards to get restaurant food vs. grocery stores wasn’t considered. I’m assuming it was due to logistics of what was most feasible to provide, but it would be helpful to address this somewhere and mention in the limitations section and it makes it harder to determine what was more appealing to the students. Including the qualitative interviews was helpful, so I’d mention this as a way you tried to address this limitation through your study design.

Figure 1

The information in this figure doesn’t seem to match the text. In the text, it’s mentioned that they received each benefit for 2 months, yet the figure says first intervention (1 month) and second intervention (1 month).

Quantitative analysis: It could also be interesting to look at which statements from the food security survey module changed. Perhaps a table showing responses to the different questions at each of the data collection points.

Line 158: Should “interviewees” be “interviewers?”

Results

Line 182: This is the first time the term CalFresh is used. Mention that this is what SNAP is called in California. Also, the information mentioned is in Table 1, not table 2.

Table 1: Remove the underlining under response options for adequate cooking materials. I also suggest changing materials to equipment to match what’s in the text.

Is there any information available on the students living situation? It would be helpful to know if they were living with family, roommates or on their own as well as on or off-campus.

Table 2: For food assistance, the number and percentages are not in the correct rows with the response options.

You may want to consider simplifying some of the response options to yes or no since the titles tell you what they are. Some examples of ways to consider wording those are below. I also mention where it would be helpful to further elaborate on what something means. This could be done in the table with a more descriptive title or as a footnote.

Unable to access food benefits (this could also use some additional elaboration on what you mean by this because it sounds like they couldn’t access the benefits at all but this doesn’t seem to be the case)

Yes

No

Unused Benefits (or Used All Benefits)

Yes

No

New Food Assistance in the Last 60 Days (this also needs some additional clarification. Does this mean they started food assistance during the study period?)

Yes

No

Missing

First Time Using Grubhub

Yes

No

Missing

Timing of Benefits Mattered (What does this mean?)

Yes

No

Unclear

Shared Grocery Store Benefits

Yes

No

Unsure

Shared Grubhub Benefits

Yes

No

The sharing of food total under grubhub is confusing. If you want to report how many used both vs. one benefit, this seems like it should be listed on its own. You could also consider combining the benefits to be something like below

Shared Benefits

Grocery Store

Grubhub

Both Grocery Store and Grub Hub

Neither

Table 3

Capitalize each of the words in the title to be consistent with the other tables.

Some of the quotes seem to include a participant ID at the end in parentheses while others don’t. Make sure this is consistent among all the quotes.  

Lines 191-192: What does it mean that they missed their second benefit?

Discussion

Lines: 284-286: What about SEBTC? This has been shown to improve food security status in low-income children.

https://pubmed.ncbi.nlm.nih.gov/29592869/

https://pubmed.ncbi.nlm.nih.gov/28109420/

https://pubmed.ncbi.nlm.nih.gov/28017594/

Line 295: I suggest using a different term than “emergency meal” to describe the grubhub meals as the students were given gift cards to purchase both types of foods.

Lines 213-215: Also include the number and percentage of participants who improved with both the grocery store and grubhub benefits. It’s unclear whether the same students saw improvement under both conditions or if different students benefited from different conditions.

Lines 291-293: This sentence seems like it would belong in the discussion section and could also be elaborated on. It was mentioned in lines 340-342 that some students received some benefits during the summer. Students may have had changes in living or employment situations that could have impacted their food security status. Moving home with family or getting a summer job could influence their ability to acquire food, for example.  

Lines 330-331: Add to this that the students were all from a single university

Conclusions

Lines 344-345: Because this was such as small sample, you may want to add that this was for a sample of college students from CSUEB.

Lines 345-350: With only 3 students from one university saying they preferred receiving the restaurant delivery benefits, I don’t think the findings support the conclusion being made. I think that a more appropriate recommendation would be to do further research on restaurant delivery service benefits for college students. You could also provide recommendations for what this research might look like based on some of the findings and limitations in this study.

Author Response

(The authors gave the same response as above.)

Reviewer 3 Report

This is a very important public health issue for college students.

There should be a justification (review of psychometrics) for using the USDA 10 item Food Security Survey in college students. 

There should also be a justification for limiting this feasibility to only marginally food insecure students (why exclude low and very low food secure students?). 

For "...this study identified the cost associated with improving food security status, which may be more cost-effective than some other interventions," there is no analysis mentioned.  Please include the cost analysis in the results. 

Please proofread for errors in grammar.

Author Response

(The authors gave the same response as above.)

Round 2

Reviewer 3 Report

Authors have addressed all concerns.